# Mapping Eastern (EEE) and Venezuelan Equine Encephalitides (VEE) among Equines Using Geographical Information Systems, Colombia, 2008–2019

**DOI:** 10.3390/v15030707

**Published:** 2023-03-08

**Authors:** D. Katterine Bonilla-Aldana, Christian David Bonilla Carvajal, Emilly Moreno-Ramos, Joshuan J. Barboza, Alfonso J. Rodriguez-Morales

**Affiliations:** 1Research Unit, Universidad Continental, Huancayo 12000, Peru; 2Faculty of Veterinary Medicine, Fundación Universitaria Autónoma de las Américas, Pereira 660003, Colombia; 3Faculties of Veterinary Medicina and Medicine, Institución Universitaria Visión de las Américas, Pereira 660003, Colombia; 4Universidad San Ignacio de Loyola, Lima 13008, Peru; 5Vicerrectorado de Investigacion, Universidad Norbert Wiener, Lima 15046, Peru; 6Faculty of Health Sciences, Universidad Científica del Sur, Lima 15067, Peru; 7Grupo de Investigación Biomedicina, Faculty of Medicine, Fundación Universitaria Autónoma de las Américas, Pereira 660003, Colombia; 8Gilbert and Rose-Marie Chagoury School of Medicine, Lebanese American University, Beirut 1102, Lebanon

**Keywords:** eastern equine encephalitis virus (EEEV), infection, Venezuelan equine encephalitis (VEEV), GIS, geographic information systems, equines, epidemiology

## Abstract

Introduction: Eastern equine encephalitis virus (EEEV) and Venezuelan equine encephalitis virus (VEEV) viruses are zoonotic pathogens affecting humans, particularly equines. These neuroarboviruses compromise the central nervous system and can be fatal in different hosts. Both have significantly influenced Colombia; however, few studies analyse its behaviour, and none develop maps using geographic information systems to characterise it. Objective: To describe the temporal-spatial distribution of those viruses in Colombia between 2008 and 2019. Methods: Retrospective cross-sectional descriptive study, based on weekly reports by municipalities of the ICA, of the surveillance of both arboviruses in equines, in Colombia, from 2008 to 2019. The data were converted into databases in Microsoft Access 365^®^, and multiple epidemiological maps were generated with the Kosmo RC1^®^3.0 software coupled to shape files of all municipalities in the country. Results: In the study period, 96 cases of EEE and 70 of VEE were reported, with 58% of EEE cases occurring in 2016 and 20% of EEV cases in 2013. The most affected municipalities for EEE corresponded to the department of Casanare: Yopal (20), Aguazul (16), and Tauramena (10). In total, 40 municipalities in the country reported ≥1 case of EEE. Conclusions: The maps allow a quick appreciation of groups of neighbouring municipalities in different departments (1° political division) and regions of the country affected by those viruses, which helps consider the expansion of the disease associated with mobility and transport of equines between other municipalities, also including international borders, such as is the case with Venezuela. In that country, especially for EEV, municipalities in the department of Cesar are bordering and at risk for that arboviral infection. there is a high risk of equine encephalitis outbreaks, especially for VEE. This poses a risk also, for municipalities in the department of Cesar, bordering with Venezuela.

## 1. Introduction

Vector-borne diseases remain a significant public health problem in tropical and subtropical regions, especially those of viral aetiology. Beyond dengue, chikungunya, and Zika, other particularly zoonotic arboviruses, such as the case of equine encephalitides, are especially relevant in the Americas [1]. Venezuelan Equine Encephalitis virus (VEEV) has been especially important after the large outbreak across the Venezuelan-Colombian border in 1995; this led to approximately ~75,000 cases in humans, ~4% with neurological consequences and a case fatality rate of 0.4% (4 per 1000 cases), just in the La Guajira State of Colombia. In that area, ~50,000 horses were infected, with 4000 fatal outcomes (~8%) [2,3].

The Venezuelan equine encephalitis virus causes Venezuelan equine encephalomyelitis. Mosquito vectors, such as *Culex* spp. (involved in the enzootic cycle), *Ochlerotatus*, and *Psorophora* spp., transmit the virus. *Culex* serves as amplifying vector, and *Ochleratatus* and *Psorophora* as bridge vectors. It mainly affects different species of equids, humans, and wild animals. However, the enzootic cycle of VEEV is complex and involves many other species of mammals. Some of the enzootic viral species within the alphavirus genus are Mosso das Pedras virus, Everglades virus, Mucambo virus, Tonate virus, Pixuna virus, Cabassou virus, and Río Negro virus, among others [1].

Besides VEEV, a related alphavirus is eastern equine encephalitis (EEE). EEE is one of the most severe arboviral encephalitides that occurs in different countries in the Americas. It has been reported in northern and southern countries, including Canada, the United States of America (USA), Belize, Costa Rica, Panama, Colombia, Venezuela, Ecuador, Guyana, Suriname, Brazil, and Bolivia [1].

EEE is considered an emerging viral disease and is showing a steady increase in incidence in a broader population. In the USA, six to eight cases are reported annually, predominantly between May and October, primarily in Florida, Georgia, Maryland, Wisconsin, and New Jersey. This virus has also been considered a potential bioterrorism weapon, given its airborne transmission. According to the Centers for Disease Control and Prevention (CDC), EEEV and VEEV are considered bioterrorism agents in category B, the second highest priority pathogens, including those that are moderately easy to spread, led to moderate fatality rates and low mortality rates, and require specific enhancements of CDC’s diagnostic capacity and enhanced disease surveillance (www.cdc.gov, accessed on 1 October 2022). The fatality rate of EEE is 30%, with neurological sequelae observed in approximately 50% of survivors [4].

EEEV virus was initially isolated from dead horses in Delaware, Maryland, and Virginia in 1933. However, it continues to circulate among *Culiseta melanura* mosquitoes and birds that serve as amplifying hosts in freshwater hardwood swamps [5]. Notably, *C. melanura* feeds almost exclusively on birds. Hence, the spread of the EEE virus to humans and other mammalian hosts (e.g., horses) requires invertebrate hosts such as *Aedes*, *Coquillettidia*, or *Culex* that feed on both birds and mammals [6]. *Culex pedroi* (enzootic cycle) and *Aedes taeniorhynchus* (epizootic cycle) are considered the main vectors of EEEV [1,7,8]. In the eastern United States, especially following the 2019 season, New England states (north of New Jersey) were also heavily affected by the virus [9].

Following the 1967–68 epizootic of VEE in areas of low and moderate seasonal rainfall in Colombia, a strain of the VEE virus was sought in the high-rainfall region of the Pacific lowlands, where previous serological studies had shown past VEE activity. VEE virus in humans without indication of clinical disease manifests in epidemic form. A total of 20 sentinel hamsters, in two groups, were exposed for 2-week periods, one in July and one in August and September 1969, along the margins of a grassy freshwater swamp area 50 km inland from the port of Tumaco, department of Nariño, near the north Ecuadorian border. In Colombia, departments are the first political divisions of the territory (e.g., analogous to states in the United States). One hamster developed an infection from the VEE virus, and two others from the EEE virus [8,10,11]. EEE virus isolates were the first from Colombia, the first from the west coast of South America, and the first from sentinel hamsters [12]. The circulation of the EEE virus in Colombia has been reported since 1957, and species such as *Culex bidens* were suspected to be this virus’s vector during an epizootic in Argentina in 1988. In 2016, *C. bidens* was identified in the municipality of Santa Cruz de Lorica, Córdoba, on the north Caribbean coast of Colombia [13].

Both viral pathologies (EEE and VEE) can cause severe compromise of the central nervous system and death, both in animals and in humans, with the ability to persist over time due to having multiple vertebrate hosts. Despite all of the above, there are few studies on VEE [14,15,16,17,18] and even more on EEE in Colombia [19,20]. The objective of the current study is to characterise the temporal-spatial distribution of EEE and VEE in Colombia between 2008 and 2019.

## 2. Methods

### 2.1. Type of Study

A retrospective cross-sectional descriptive observational study was conducted to assess the incidence of EEE and VEE in horses in Colombia. Their incidence per year for the Colombian departments was estimated, and epidemiological maps of the diseases were developed covering the years 2008 to 2019. Additionally, for 2016 to 2019, the annual number of equines per department was obtained. For that period, the incidence rate of EEE and VEE was estimated by calculating the number of cases per 100,000 equines per department per year, confirmed by RT-PCR, by the accredited ICA central reference laboratories.

This is a study of secondary sources where bulletins from the Columbian Agricultural Institute (ICA) were used to obtain local sociodemographic data and records of disease occurrences and their locations, including disease diagnosis. In addition, the total number of cases and their regional location were taken for the geographical characterisation of the incidence of VEE-EEE in the departments of Colombia.

For the respective geographical characterisation, the free access software Kosmo 3.1 was used, which comes with preloaded tools for the geographical analyses. For this investigation, West Bogotá (West Bogotá) (Magna SIRGAS) was used as a reference system (EPSG 21896) of the geographic coordinate system. The mapping was carried out (at a scale of 1: 1,365,207) for which two types of layers were used; the first layer was the departments, to which colours of greater and lesser intensity were added, defined by the epidemiological data layer, classified by ranges established by quartiles, which made it possible to differentiate the areas with a higher incidence of the disease from those with a lower incidence.

### 2.2. Population

Horses from Colombia between the periods 2008–2019 that were diagnosed positive for EEE and VEE, according to the ICA databases.

### 2.3. Sample

The sample was a census taken from the ICA databases, from which registered data will be taken from cases of diagnosis of EEE and VEE in horses in Colombia between 2008 and 2019, confirmed by molecular testing, RT-PCR, by the accredited ICA central reference laboratories.

### 2.4. Analyses

Descriptive statistics were performed on Stata IC^®^ version 14. The summary of cases per year and departments and the median number and their interquartile (IQR) cases per year for specific periods were calculated.

## 3. Results

In the study period (2008–2019), 96 cases of EEE (median 2 cases/year, IQR 0–7) and 70 of VEE were reported (median 6 cases/year, IQR 3–8), with 58% of EEE cases occurring in 2016 (Figure 1) and 20% of VEE cases in 2013 (Figure 2).

The most affected municipalities for EEE corresponded to the department of Casanare: Yopal (20), Aguazul (16) and Tauramena (10) (Figure 3, Figure 4, Figure 5, Figure 6 and Figure 7). A total of 40 municipalities in the country reported ≥1 case of EEE in the period (Figure 3). For VEE, the cases were less concentrated, with Montelíbano, Córdoba being the municipality with the most reported cases (4) (Figure 8, Figure 9, Figure 10 and Figure 11). A total of 44 municipalities reported ≥1 case of VEE in the period (Figure 8).

During the first six years (2008–2013), a smaller number of cases and affected municipalities was observed for EEV (Figure 3) compared with the following six years (2014–2019) (Figure 3, Figure 4, Figure 5, Figure 6 and Figure 7). On the other hand, the highest concentration of cases and affected municipalities in VEE occurred from 2013 to 2016 (Figure 8, Figure 9, Figure 10 and Figure 11).

In 2016, there was a concentration of cases in the neighbouring municipalities in the Casanare department, amounting to 93% of the annual cases for that year (Figure 5). In 2017, municipalities such as Cumaribo, La Primavera, and Arauca, which border Venezuela, reported cases of EEE (Figure 6). In 2013, 2014, and 2016, a similar situation occurred for Tibú, Agustin Codazzi, Becerril, and Curumaní municipalities, which also border with that country and reported cases of VEE (Figure 9, Figure 10 and Figure 11). In addition, one capital municipality reported cases of VEE in 2014 and 2016, Valledupar.

For 2016–2019, the highest incidence rates of VEE (cases per 100,000 animals) were observed in 2016, 0.829 cases per 100,000 animals (8.29 cases per 1,000,000 animals) (Table 1).

In 2016, the department with the highest incidence rate was Cesar (14.165 cases per 100,000 animals). However, the national rates decreased in the following years to 0.346 cases per 100,000 animals (3.46 cases per 1,000,000 animals) in 2017, to 0.269 cases per 100,000 animals (2.69 cases per 1,000,000 animals) in 2018, and 0 cases per 100,000 animals in 2019 (Table 1). For 2017, the department with the highest incidence rate was Cordoba (1.842 cases per 100,000 animals). In 2018, at Chocó, the incidence rate was 18.150 cases per 100,000 animals. In 2019 no cases were reported (Table 1).

For 2016–2019, the highest incidence rates of EEE (cases per 100,000 animals) were observed in 2016, 3.870 cases per 100,000 animals (38.7 cases per 1,000,000 animals) (Table 2). In 2016, the department with the highest incidence rate was Casanare (62.999 cases per 100,000 animals). However, the national rates decreased in the following years to 1.176 cases per 100,000 animals (11.76 cases per 1,000,000 animals) in 2017, to 0.269 cases per 100,000 animals (2.69 cases per 1,000,000 animals) in 2018, and 0 cases per 100,000 animals in 2019 (Table 2). For 2017, the department with the highest incidence rate was Vichada (56.417 cases per 100,000 animals). In 2018, at Guaviare, the incidence rate was 9.708 cases per 100,000 animals. In 2019 only Meta reported cases with an incidence rate of 3.159 per 100,000 animals (Table 2). On the border with Venezuela, the Cesar department (Figure 11) had 2016 high incidence rates of VEE (Table 3).

## 4. Discussion

Zoonotic arboviruses may be considered even more complex than those that are only anthroponotic or are predominantly anthroponotic [21]. Equine encephalitides are suitable examples of that. These arboviruses include many flavi and alphaviruses and have a wide range of animal hosts and humans. Birds and horses, specifically, are key animals in their enzootic cycles [21]. In the case of horses, these are animals that may serve as sentinels for these vector-borne diseases, as well as for others, mainly due to the ease of access to them, given their proximity to human beings. Regarding birds, the fact that many of them are migratory is a matter of concern for international spreading in countries related to migration routes.

Equine encephalitides such as EEE and VEE are still significant zoonotic arboviral threats in Colombia, Venezuela, and other countries of Latin America and probably the Caribbean islands [1,22,23,24]. In the current study, a wide distribution during the study period was observed in both alphavirus infections. Nevertheless, with higher variations in EEE (0 to 56 cases in a year), reaching a peak of cases in 2016, without significant variations in the number of cases per year for VEE (0 to 14 cases per year) were observed.

As expected, circulation in municipalities bordering Venezuela was observed in both infections, which may create concern given the sanitary crisis in the country. Over the last few years, the sanitary crisis in Venezuela has implied the reemergence of multiple infectious diseases in humans, animals, zoonotic diseases, and a decrease in vaccination and control programmes, which have led to a complex situation. That may be related to vaccination problems and the critical situation and especially given the healthcare crisis in Venezuela [25], where there are no vaccination programmes for animals or humans (lack of immunisation for measles, mumps, yellow fever and multiple other vaccine-preventable diseases), and where numerous infectious diseases have re-emerged, including foot-and-mouth disease that was reintroduced to Colombia from Venezuela [25,26,27,28,29,30,31]. Furthermore, the border crossing of human beings from Venezuela to Colombia has been documented among patients with imported infections, that in some cases, try to reach other borders, such as the one between Colombia and Ecuador in the South or between Colombia and Panama trying to reach the “American dream” after crossing the Darién gap [32,33,34,35]. Although humans are not important reservoir hosts for VEE or EEE, they may cross borders infected and may be diagnosed in other territories and countries.

Additionally, although they may not be a suitable source for transmission, they may impact the health system, care, and disease burden in destination areas. Domestic, farm and wild animals, including equines and birds, may cross the borders (in areas without border controls), sometimes even with their owners, with multiple implications, including EEE and VEE. Therefore, more controls and regulations regarding equines, including vaccination, should be considered, as they may add an additional burden to the healthcare systems of destination countries. In the case of VEE, live-attenuated vaccines have been used in the U.S. military and laboratory workers, and formalin-inactivated vaccines are available for use in horses [36]. One such live-attenuated vaccine is TC-83, developed initially by the U.S. army for vaccine use [37]. TC-83 was created by serially passaging the Trinidad Donkey VEEV strain in guinea pig heart cells [38]. In recent years, new vaccines for horses have been investigated [39], but even a phase 1, open-label, dose-escalation, randomised clinical trial, assessing the safety and immunogenicity of a trivalent virus-like particle vaccine against Western, Eastern, and Venezuelan Equine Encephalitis viruses in humans found a favourable safety profile and neutralising antibody responses including high response against VEE and EE. Along with pressing public health needs, evidence supports further evaluation of the vaccine in advanced-phase clinical trials [40]. In this context, especially vaccination against VEE plays a key role in control. However, the enhancement in the vaccination programme against eastern and Venezuelan equine encephalitis viruses should be performed not only in Colombia but across the border to Venezuela. Given the current sociopolitical situation, cooperation between both governments should be held in order to improve the disease control and vaccination programmes in both countries and specifically on the border. Such arboviral diseases may add an additional burden to the healthcare systems of destination countries.

In the case of EEE, this neuroarbovirus has been scarcely investigated in Colombia, with municipalities of the department of Casanare, such as Yopal, Aguazul, San Luis de Palenque, Trinidad, Orocué, Mani, and Tauramena, affected in 2016. Additionally, in 2017, many municipalities of the eastern plains or lowlands presented cases of EEE in the departments of Meta, Guaviare, Arauca, and Vichada. The maps developed show a clear circulation corridor that goes from the municipality of La Primavera (Vichada) on the border with Venezuela, passing through Cumaribo (Vichada), San José del Guaviare (Guaviare), Puerto Rico (Meta) and Puerto Lleras (Meta) (Figure 6). Surveillance for eastern and Venezuelan equine encephalitis viruses needs to be increased in the country in humans and horses, as well as studies in birds are also crucial for understanding its transmission. In addition, entomological surveillance is critical in studying such cycles [41], including the study of the mosquito’s host preference, which may be helpful in understanding vector feeding [16].

VEE is a persistent zoonosis in Colombia through long periods of apparent epidemiological silence in horses and humans [41,42,43]. In addition to equine studies, it is essential to know the behaviour of rodents as well as bats [24,44], which could have a potential role as dispersing hosts of this virus. Therefore, equine surveillance is helpful as a predictive or sentinel indicator to prevent human cases. Similarly, EEE, to date, has not been reported in Colombia in humans. Nevertheless, a member of the eastern equine encephalitis virus (EEEV) complex, Madariaga virus (MADV) is currently registered in multiple countries of Latin America [45], such as Brazil, Panama [46], Haiti [47], Venezuela [48], among others and in addition to Argentina [49], where it was described.

## 5. Conclusions

The developed GIS-based maps generated have allowed us to quickly appreciate groups of neighbouring municipalities in different departments and regions of the country affected by EEE and VEE, helps to appreciate the expansion of the disease associated with mobility and transportation of equines between different municipalities, also including international borders, as it is the case with Venezuela, especially for VEE, with municipalities of the department of Cesar among others. Furthermore, these results have implications for public health in planning and surveillance, especially in those municipalities close to those historically affected in the study period. Future assessments of equine encephalitides should extend to other potential threats, including MADV, Western Equine Encephalitis virus, West Nile Virus, Saint Louis encephalitis virus, reported in Brazil [18,50,51,52], Rocio virus [53,54], among others, that may even be considered neglected arboviruses in Latin America [55].

## Figures and Tables

**Figure 1 viruses-15-00707-f001:**
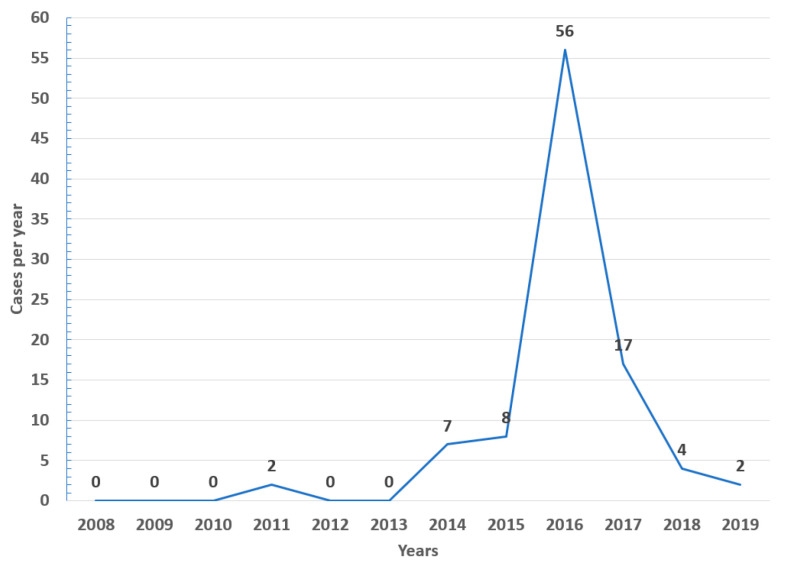
Yearly cases of EEE among equines in Colombia from 2008 to 2019.

**Figure 2 viruses-15-00707-f002:**
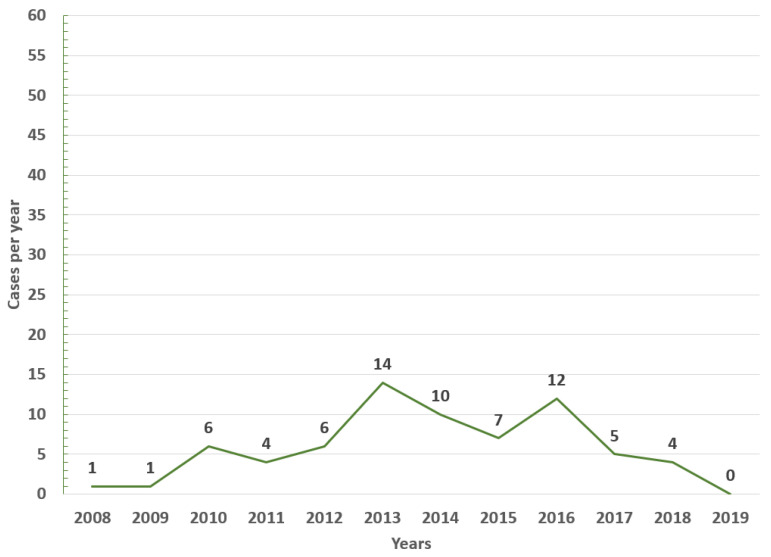
Yearly cases of VEE among equines in Colombia from 2008 to 2019.

**Figure 3 viruses-15-00707-f003:**
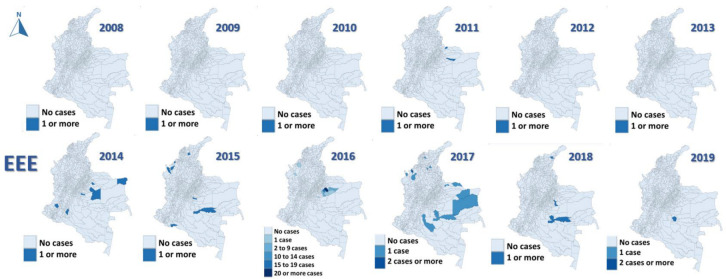
Distribution maps of EEE among equines, Colombia, 2008–2019.

**Figure 4 viruses-15-00707-f004:**
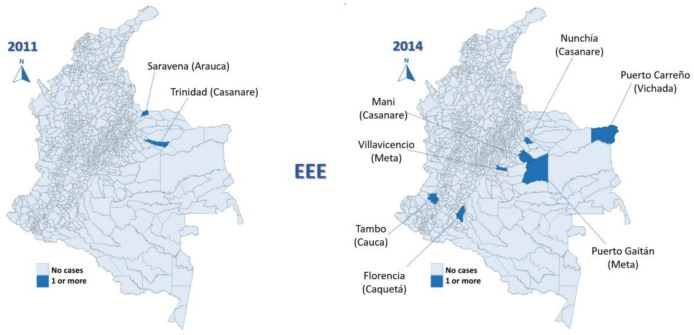
EEE among equines in affected municipalities of Colombia, 2011 and 2014.

**Figure 5 viruses-15-00707-f005:**
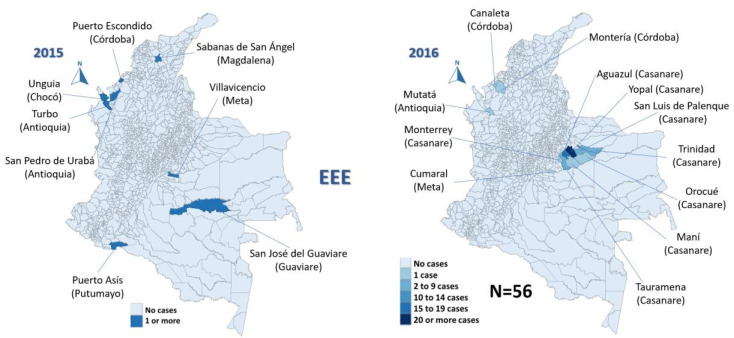
EEE among equines in affected municipalities of Colombia, 2015 and 2016.

**Figure 6 viruses-15-00707-f006:**
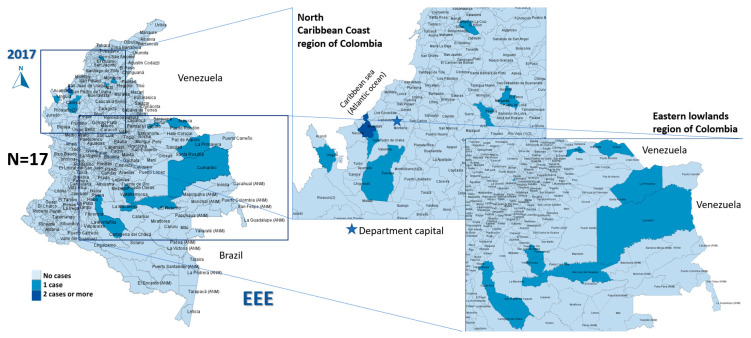
EEE among equines in affected municipalities of Colombia, 2017.

**Figure 7 viruses-15-00707-f007:**
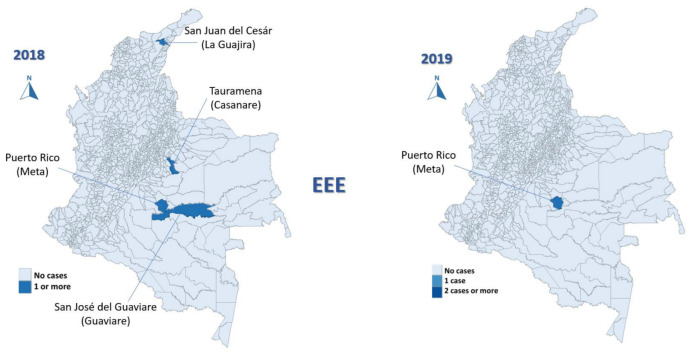
EEE among equines in affected municipalities of Colombia, 2018 and 2019.

**Figure 8 viruses-15-00707-f008:**
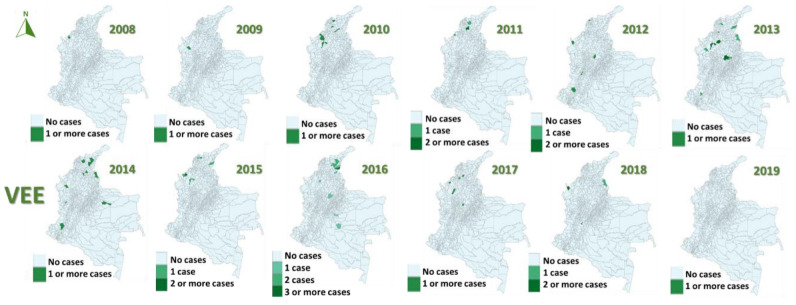
Distribution maps of VEE among equines, Colombia, 2008–2019.

**Figure 9 viruses-15-00707-f009:**
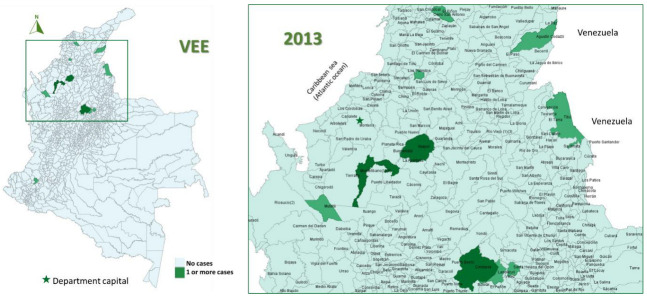
VEE among equines in affected municipalities of Colombia, 2013.

**Figure 10 viruses-15-00707-f010:**
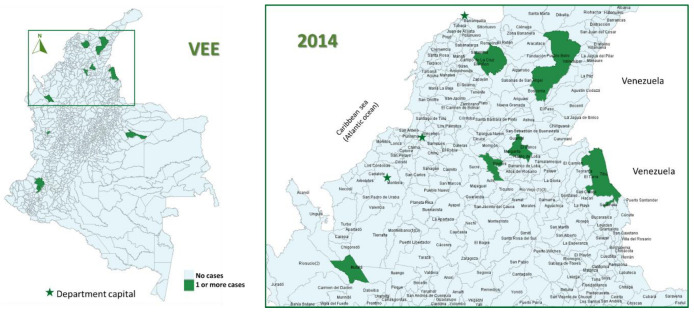
VEE among equines in affected municipalities of Colombia, 2014.

**Figure 11 viruses-15-00707-f011:**
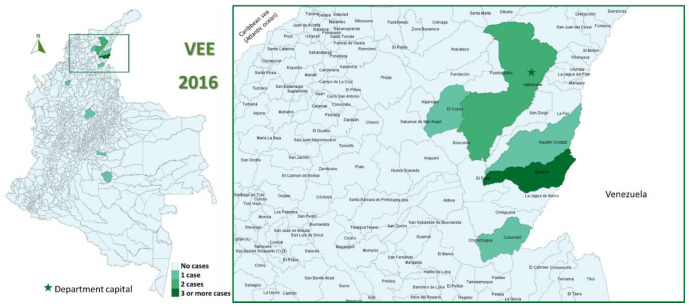
VEE among equines in affected municipalities of Colombia, 2016.

**Table 1 viruses-15-00707-t001:** VEE cases, equine population, and equine VEE annual incidence per department, Colombia, 2016–2019.

Department	Cases per Year				Population per Year				Annual Incidence (Cases/100,000 Animals)			
	2016	2017	2018	2019	2016	2017	2018	2019	2016	2017	2018	2019
Amazonas	0	0	0	0	0	52	50	65	0.000	0.000	0.000	0.000
Antioquia	0	1	0	0	153,970	165,213	178,018	221,463	0.000	0.605	0.000	0.000
Arauca	0	0	0	0	49,800	55,402	58,170	52,053	0.000	0.000	0.000	0.000
Atlantico	0	0	0	0	12,700	10,663	9942	199	0.000	0.000	0.000	0.000
Bolivar	0	0	0	0	53,808	53,356	57,071	10,101	0.000	0.000	0.000	0.000
Boyaca	0	0	0	0	48,268	48,268	46,868	57,386	0.000	0.000	0.000	0.000
Caldas	0	0	0	0	8731	18,956	21,110	52,926	0.000	0.000	0.000	0.000
Caqueta	0	0	0	0	48,349	48,349	54,563	41,014	0.000	0.000	0.000	0.000
Casanare	0	0	0	0	82,541	99,208	90,049	63,639	0.000	0.000	0.000	0.000
Cauca	0	0	0	0	81,764	82,483	86,834	97,080	0.000	0.000	0.000	0.000
Cesar	8	0	0	0	56,477	58,437	61,091	89,256	14.165	0.000	0.000	0.000
Choco	0	0	2	0	11,359	10,578	11,019	60,996	0.000	0.000	18.150	0.000
Cordoba	1	2	0	0	106,829	108,604	115,473	11,069	0.936	1.842	0.000	0.000
Cundinamarca	0	0	0	0	109,077	109,077	108,455	112,805	0.000	0.000	0.000	0.000
Guainia	0	0	0	0	12,190	12,190	154	110,779	0.000	0.000	0.000	0.000
Guaviare	0	0	0	0	11,130	8561	10,301	105	0.000	0.000	0.000	0.000
Huila	0	0	0	0	64,841	56,310	56,310	11,015	0.000	0.000	0.000	0.000
La-Guajira	0	0	0	0	21,276	14,025	21,038	45,603	0.000	0.000	0.000	0.000
Magdalena	0	1	0	0	62,544	56,344	60,289	21,050	0.000	1.775	0.000	0.000
Meta	2	0	0	0	66,037	72,184	77,415	63,307	3.029	0.000	0.000	0.000
Narino	0	0	0	0	54,209	55,305	52,323	79,998	0.000	0.000	0.000	0.000
Norte de Santander	0	0	1	0	26,501	17,548	24,537	47,661	0.000	0.000	4.075	0.000
Putumayo	0	0	0	0	11,596	11,982	11,614	26,621	0.000	0.000	0.000	0.000
Quindio	0	0	0	0	3123	6995	6995	13,505	0.000	0.000	0.000	0.000
Risaralda	0	0	0	0	9100	11,670	6267	6383	0.000	0.000	0.000	0.000
San Andres	0	0	0	0	158	155	166	5691	0.000	0.000	0.000	0.000
Santander	1	0	0	0	77,200	57,797	58,425	59,197	1.295	0.000	0.000	0.000
Sucre	0	1	0	0	56,879	54,371	54,703	62,908	0.000	1.839	0.000	0.000
Tolima	0	0	1	0	118,214	118,590	118,590	146,631	0.000	0.000	0.843	0.000
Valle	0	0	0	0	19,961	19,777	20,747	24,315	0.000	0.000	0.000	0.000
Vaupes	0	0	0	0	28	21	91	40	0.000	0.000	0.000	0.000
Vichada	0	0	0	0	8436	3545	7544	8153	0.000	0.000	0.000	0.000
**Total**	**12**	**5**	**4**	**0**	**1,447,096**	**1,446,016**	**1,486,222**	**1,603,014**	**0.829**	**0.346**	**0.269**	**0.000**

**Table 2 viruses-15-00707-t002:** EEE cases, equine population, and equine EEE annual incidence per department, Colombia, 2016–2019.

Department	Cases per Year				Population per Year				Annual Incidence (Cases/100,000 Animals)			
	2016	2017	2018	2019	2016	2017	2018	2019	2016	2017	2018	2019
Amazonas	0	0	0	0	0	52	50	65	0.000	0.000	0.000	0.000
Antioquia	1	3	0	0	153,970	165,213	178,018	221,463	0.649	1.816	0.000	0.000
Arauca	0	2	0	0	49,800	55,402	58,170	52,053	0.000	3.610	0.000	0.000
Atlantico	0	0	0	0	12,700	10,663	9942	199	0.000	0.000	0.000	0.000
Bolivar	0	1	0	0	53,808	53,356	57,071	10,101	0.000	1.874	0.000	0.000
Boyaca	0	1	0	0	48,268	48,268	46,868	57,386	0.000	2.072	0.000	0.000
Caldas	0	0	0	0	8731	18,956	21,110	52,926	0.000	0.000	0.000	0.000
Caqueta	0	1	0	0	48,349	48,349	54,563	41,014	0.000	2.068	0.000	0.000
Casanare	52	0	1	0	82,541	99,208	90,049	63,639	62.999	0.000	1.111	0.000
Cauca	0	0	0	0	81,764	82,483	86,834	97,080	0.000	0.000	0.000	0.000
Cesar	0	0	0	0	56,477	58,437	61,091	89,256	0.000	0.000	0.000	0.000
Choco	0	1	0	0	11,359	10,578	11,019	60,996	0.000	9.454	0.000	0.000
Cordoba	2	1	0	0	106,829	108,604	115,473	11,069	1.872	0.921	0.000	0.000
Cundinamarca	0	0	0	0	109,077	109,077	108,455	112,805	0.000	0.000	0.000	0.000
Guainia	0	0	0	0	12,190	12,190	154	110,779	0.000	0.000	0.000	0.000
Guaviare	0	1	1	0	11,130	8561	10,301	105	0.000	11.681	9.708	0.000
Huila	0	0	0	0	64,841	56,310	56,310	11,015	0.000	0.000	0.000	0.000
La-Guajira	0	0	1	0	21,276	14,025	21,038	45,603	0.000	0.000	4.753	0.000
Magdalena	0	2	0	0	62,544	56,344	60,289	21,050	0.000	3.550	0.000	0.000
Meta	1	2	1	2	66,037	72,184	77,415	63,307	1.514	2.771	1.292	3.159
Narino	0	0	0	0	54,209	55,305	52,323	79,998	0.000	0.000	0.000	0.000
Norte de Santander	0	0	0	0	26,501	17,548	24,537	47,661	0.000	0.000	0.000	0.000
Putumayo	0	0	0	0	11,596	11,982	11,614	26,621	0.000	0.000	0.000	0.000
Quindio	0	0	0	0	3123	6995	6995	13,505	0.000	0.000	0.000	0.000
Risaralda	0	0	0	0	9100	11,670	6267	6383	0.000	0.000	0.000	0.000
San Andres	0	0	0	0	158	155	166	5691	0.000	0.000	0.000	0.000
Santander	0	0	0	0	77,200	57,797	58,425	59,197	0.000	0.000	0.000	0.000
Sucre	0	0	0	0	56,879	54,371	54,703	62,908	0.000	0.000	0.000	0.000
Tolima	0	0	0	0	118,214	118,590	118,590	146,631	0.000	0.000	0.000	0.000
Valle	0	0	0	0	19,961	19,777	20,747	24,315	0.000	0.000	0.000	0.000
Vaupes	0	0	0	0	28	21	91	40	0.000	0.000	0.000	0.000
Vichada	0	2	0	0	8436	3545	7544	8153	0.000	56.417	0.000	0.000
**Total**	**56**	**17**	**4**	**2**	**1,447,096**	**1,446,016**	**1,486,222**	**1,603,014**	**3.870**	**1.176**	**0.269**	**0.125**

**Table 3 viruses-15-00707-t003:** VEE cases, equine population, and equine EEE annual incidence per municipalities at the department of Cesar, Colombia, 2016–2019.

Municipality	Cases per Year				Population per Year				Annual Incidence (Cases/100,000 Animals)			
	2016	2017	2018	2019	2016	2017	2018	2019	2016	2017	2018	2019
Aguachica	0	0	0	0	2439	2419	2443	2445	0.000	0.000	0.000	0.000
Agustín Codazzi	1	0	0	0	4020	4031	4036	4035	24.876	0.000	0.000	0.000
Astrea	0	0	0	0	2500	2797	2915	2918	0.000	0.000	0.000	0.000
Becerril	3	0	0	0	1200	1436	2171	2180	250.000	0.000	0.000	0.000
Bosconia	0	0	0	0	2300	2146	2164	2165	0.000	0.000	0.000	0.000
Chimichagua	0	0	0	0	2400	2681	2892	2893	0.000	0.000	0.000	0.000
Chiriguaná	0	0	0	0	2010	3459	4369	4370	0.000	0.000	0.000	0.000
Curumaní	1	0	0	0	2050	2604	3092	3094	48.780	0.000	0.000	0.000
El Copey	1	0	0	0	2610	2685	2456	2460	38.314	0.000	0.000	0.000
El Paso	0	0	0	0	1850	2595	3047	3050	0.000	0.000	0.000	0.000
Gamarra	0	0	0	0	1310	1229	1298	1300	0.000	0.000	0.000	0.000
González	0	0	0	0	650	580	59	62	0.000	0.000	0.000	0.000
La Gloria	0	0	0	0	2350	2337	2310	2312	0.000	0.000	0.000	0.000
La Jagua de Ibirico	0	0	0	0	1600	1450	1254	1256	0.000	0.000	0.000	0.000
La Paz	0	0	0	0	2400	2450	2635	2638	0.000	0.000	0.000	0.000
Manaure	0	0	0	0	850	850	202	205	0.000	0.000	0.000	0.000
Pailitas	0	0	0	0	1810	1432	1427	1430	0.000	0.000	0.000	0.000
Pelaya	0	0	0	0	2010	1319	1387	1390	0.000	0.000	0.000	0.000
Pueblo Bello	0	0	0	0	1270	533	427	430	0.000	0.000	0.000	0.000
Río de Oro	0	0	0	0	1660	1343	1488	1490	0.000	0.000	0.000	0.000
San Alberto	0	0	0	0	1772	1681	1665	1664	0.000	0.000	0.000	0.000
San Diego	0	0	0	0	2550	2316	2216	2114	0.000	0.000	0.000	0.000
San Martín	0	0	0	0	2400	2100	3061	3055	0.000	0.000	0.000	0.000
Tamalameque	0	0	0	0	1913	1946	1786	1790	0.000	0.000	0.000	0.000
Valledupar	2	0	0	0	8553	10,018	10,291	10,250	23.384	0.000	0.000	0.000
**Total**	**8**	**0**	**0**	**0**	**56,477**	**58,437**	**61,091**	**60,996**	**14.165**	**0.000**	**0.000**	**0.000**

## Data Availability

Available upon reasonable request.

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
