# Peer review of "Mapping Eastern (EEE) and Venezuelan Equine Encephalitides (VEE) among Equines Using Geographical Information Systems, Colombia, 2008–2019"

_viruses, 2023, doi:10.3390/v15030707_

Round 1

Reviewer 1 Report

Improvement in English is needed as there are various sections that are not readily intelligible or are not clearly laid out.

What were the criteria for making the original diagnoses of disease in humans or horses ? Were any laboratory conformations made and what were the test methods used ? Was such information indicated by the primary sources ?

Consistency is needed in the spelling of virus names (use of capitals). Refer to international nomenclature guidelines.

Under References, capitals in the titles of articles shouldn't be used (line 328, 341 etc.).

There is a need to reduce the amount of data presented and to summarize further - perhaps into a short communication, if the Journal allows for this. 

Author Response

Improvement in English is needed as there are various sections that are not readily intelligible or are not clearly laid out.

Re: We have now checked, revised and improved the manuscript.

What were the criteria for making the original diagnoses of disease in humans or horses ? Were any laboratory conformations made and what were the test methods used ? Was such information indicated by the primary sources ?

Re: The diagnosis of EEE and VEE is made by RT-PCR. Now this has been clarified in the manuscript.

Consistency is needed in the spelling of virus names (use of capitals). Refer to international nomenclature guidelines.

Re: Done. Now has been corrected.

Under References, capitals in the titles of articles shouldn't be used (line 328, 341 etc.).

Re: Done. Now has been corrected.

There is a need to reduce the amount of data presented and to summarize further - perhaps into a short communication, if the Journal allows for this.

Re: This is an Original Article, not a Short Communication, so, there is no need to do that. In fact, when we originally submitted the manuscript, it was short, and we were requested specifically from the journal to extend it to have at least 4,000 words (its body has 4,032).

Author Response

Line 44. Include ‘regions’ after sub tropical

Re: Done. Corrected.

Lines 45-46: Rewrite to “…and Zika, other particularly zoonotic arboviruses,…”

Re: Done. Corrected.

Lines 53-55: Reword to describe Culex as amplifying vector and Ochleratatus and Psorophora as bridge vectors. The authors may wish to include examples of species affected as well.

Re: Done. Corrected.

Lines 60-64: EEE (in North America) is an alphavirus not a flavivirus. Authors might consider combining this with the later chapter on vectors of EEV in South America. In line 61, replace the word ‘affects’ with ‘occurs.’

Re: Done. Both things corrected.

Lines 76-68. In the eastern United States, especially following the 2019 season, New England states (north of New Jersey) were also heavily affected by the virus. Citation would be: Armstrong, P. M., & Andreadis, T. G. (2022). Ecology and Epidemiology of Eastern Equine Encephalitis Virus in the Northeastern United States: An Historical Perspective. Journal of Medical Entomology, 59(1), 1-13.

Re: Done. Included. Reference, cited.

Line 70: change ‘bioterrorism agents/disease’ to ‘select bioterrorism agents’

Re: Done. Corrected.

Line 76: Authors are talking about multiple animal isolations so change ‘horse’ to ‘horses’

Re: Done. Corrected.

Line 86: Change ‘man’ to ‘human’

Re: Done. Corrected.

Round 2

Reviewer 1 Report

Viruses (ISSN 1999-4915) Manuscript ID   viruses-2041784

Mapping Eastern (EEE) and Venezuelan Equine Encephalitides (VEE) among Equines using Geographical Information Systems, Colombia, 2008-2019

Authors

D. Katterine Bonilla-Aldana , Christian David Bonilla Carvajal , Emilly Moreno-Ramos , Joshuan J. Barboza * , Alfonso J. Rodriguez-Morales

The following are suggested spelling and grammatical corrections.

Abstract

Overuse of capitals ; ‘virus’ to be included:

Line 23   :  eastern equine encephalitis virus and Venezuelan equine encephalitis virus

Line 40. Sentence incomplete.

Keywords

Line 42: Eastern equine encephalitis virus (EEEV), Venezuelan encephalitis virus (VEEV)

Introduction

Line 50

Venezuelan encephalitis virus (VEEV)

Line 51

Venezuelan and Columbian border

Line 64

eastern equine encephalitis (EEE)

Line 74

EEEV and VEEV are considered bioterrorism agents…

Line 80

EEEV was  initially…

Line 86

the main vectors of EEEV…

Line 96&97

... (analogous to states in the USA)…

Line 119 - 123

....where bulletins from the Columbian Agricultural Institute were used to obtain local sociodemographic  data and records of disease occurrences and there locations.

Line 137

Can further details be given as to where the tests were done, status of the labs (a central lab,; accredited ?) and what PCR methods were used (references) ?

Line 152

During the first six years (2008-2013), a smaller number of cases and affected municipalities was observed for EEV (Figure 3) compared with the following six years....

Line 156

In 2016, there was a concentration of cases in the neighbouring municipalities in the Casanare department, amounting to 93% of the annual cases for that year.

Figure 1

Line 162 : Yearly cases of EEE among equines in Colombia from 2008-2019.

There is a duplication of the graphs (both Figures 1 and 2)

All the figure legends should be below the graphs.

Lien 247:

.. given the healthcare crisis in the country

Line 252

…are no vaccination programmes in animals or humans

Line 253: infectious diseases have re-emerged....

Line 259

..after crossing the Darién Gap..

Line 263

.. they may add an additional burden to the healthcare systems of destination countries

Line 274

..western eastern and Venezuelan  equine encephalitis viruses

Line 279

..improvement in the vaccination program...

Line 301

of the eastern equine encephalitis virus (EEEV) complex

Line 307

…the GIS-based maps generated, have allowed us to..

Line 309

...helps to appreciate the expansion of the disease ..

Line 315

..including diseases caused by western equine encephalitis virus, West Nile virus, Saint Louis encephalitis virus....

Is it possible to reduce the number of figures ? Could the EEE and VEE line graphs be combined ?

Can Fig.  4 to 7 and 9 to 11 be left out ? The editors will need to indicate if sufficient space would be available to accommodate all of them .

The figure legends should be placed below the graphs.

Author Response

Dear reviewer. 

The following are suggested spelling and grammatical corrections.

Re: Thanks. We agree on that.

Abstract

Overuse of capitals ; ‘virus’ to be included:

Re: Changed.

Line 23   :  eastern equine encephalitis virus and Venezuelan equine encephalitis virus

Re: first place abbreviations were used.

Line 40. Sentence incomplete.

Re: Corrected.

Keywords

Line 42: Eastern equine encephalitis virus (EEEV), Venezuelan encephalitis virus (VEEV)

Re: Done. Changed.

Introduction

Line 50

Venezuelan encephalitis virus (VEEV)

Re: Done. Changed.

Line 51

Venezuelan and Columbian border

Re: Done. Adjusted.

Line 64

eastern equine encephalitis (EEE)

Re: Done. Adjusted.

Line 74

EEEV and VEEV are considered bioterrorism agents…

Re: Done. Adjusted.

Line 80

EEEV was  initially…

Re: Done. Changed.

Line 86

the main vectors of EEEV…

Re: Done. Changed.

Line 96&97

... (analogous to states in the USA)…

Re: Done. Changed.

Line 119 - 123

....where bulletins from the Columbian Agricultural Institute were used to obtain local sociodemographic  data and records of disease occurrences and there locations.

Re: Done. Changed.

Line 137

Can further details be given as to where the tests were done, status of the labs (a central lab,; accredited ?) and what PCR methods were used (references) ?

Re: Done. Adjusted.

Line 152

During the first six years (2008-2013), a smaller number of cases and affected municipalities was observed for EEV (Figure 3) compared with the following six years....

Re: Done. Adjusted.

Line 156

In 2016, there was a concentration of cases in the neighbouring municipalities in the Casanare department, amounting to 93% of the annual cases for that year.

Re: Done. Adjusted.

Figure 1

Line 162 : Yearly cases of EEE among equines in Colombia from 2008-2019.

Re: Done. Adjusted.

There is a duplication of the graphs (both Figures 1 and 2)

Re: There is no duplication. Figure 1 is EEE, Figure 2 is VEE. They are different.

All the figure legends should be below the graphs.

Re: Legends are below, titles up.

Lien 247:

.. given the healthcare crisis in the country

Re: Done. Adjusted.

Line 252

…are no vaccination programmes in animals or humans

Re: Done. Adjusted.

Line 253: infectious diseases have re-emerged....

Re: Done. Adjusted.

Line 259

..after crossing the Darién Gap..

Re: Done. Adjusted.

Line 263

.. they may add an additional burden to the healthcare systems of destination countries

Re: Done. Adjusted.

Line 274

..western eastern and Venezuelan  equine encephalitis viruses

Re: Done. Adjusted.

Line 279

..improvement in the vaccination program...

Re: Done. Adjusted.

Line 301

of the eastern equine encephalitis virus (EEEV) complex

Re: Done. Adjusted.

Line 307

…the GIS-based maps generated, have allowed us to..

Re: Done. Changed.

Line 309

...helps to appreciate the expansion of the disease ..

Re: Done. Changed.

Line 315

..including diseases caused by western equine encephalitis virus, West Nile virus, Saint Louis encephalitis virus....

Re: Done. Changed.

Is it possible to reduce the number of figures ? Could the EEE and VEE line graphs be combined ?

Re: If not mandatory we prefer to keep as separate for better illustration.

Can Fig.  4 to 7 and 9 to 11 be left out ? The editors will need to indicate if sufficient space would be available to accommodate all of them .

Re: Initially we have a shorter article, but it was requested by the editorial office to make longer. Now is a full-length article, online, so no problem regarding extension.

The figure legends should be placed below the graphs.

Re: They are below.
